# ‘140R’ Rootstock Regulates Resveratrol Content in ‘Cabernet Sauvignon’ Grapevine Leaves Through miRNA

**DOI:** 10.3390/plants13213057

**Published:** 2024-10-31

**Authors:** Chunmei Zhu, Zhijun Zhang, Zhiyu Liu, Wenchao Shi, Dongliang Zhang, Baolong Zhao, Junli Sun

**Affiliations:** 1Department of Horticulture, College of Agriculture, Shihezi University, Shihezi 832003, China; zhuchunmei@stu.shzu.edu.cn (C.Z.); zzj@xjau.edu.cn (Z.Z.); liuzhiyu@stu.shzu.edu.cn (Z.L.); shiwenchao@xjshzu.com (W.S.); zdl@stu.shzu.edu.cn (D.Z.); zhaobaolong@shzu.edu.cn (B.Z.); 2The Key Laboratory of Special Fruits and Vegetables Cultivation Physiology and Germplasm Resources Utilization of the Xinjiang Production and Construction, Shihezi 832003, China

**Keywords:** grape, grafting, miRNA, resveratrol

## Abstract

Grafting is important for increasing the resistance of grapevines to environmental stress, improving fruit quality, and shortening the reproductive period. In this study, ‘Cabernet Sauvignon’ (CS) grafted on the resistant rootstock 140R (CS/140R), self-grafted grapevines of the resistant rootstock 140R (140R/140R), and self-grafted grapevines of CS (CS/CS) were subjected to high-throughput sequencing; small RNA (sRNA) libraries were constructed, and miRNAs responsive to the grafting process were identified. A total of 177 known miRNAs and 267 novel miRNAs were identified. Many miRNAs responsive to the grafting process were significantly down-regulated in CS/140R leaves relative to CS/CS leaves, such as vvi-miR171c, vvi-miR171e, et al., suggesting that the expression of these miRNAs might be affected by grafting. Kyoto Encyclopedia of Genes and Genomes analysis revealed that the differentially expressed miRNAs regulated the expression of genes in the phenylpropanoid synthesis pathway. Grapevine leaves transiently overexpressing vvi-miR171c were assayed, and the expression of the target gene, *VvMYB154*, and the resveratrol content were decreased, indicating that vvi-miR171c negatively regulates the expression of *VvMYB154*. In sum, 140R increased the resveratrol content of the scion by grafting, down-regulating the expression of vvi-miR171c. These results provide new information that will aid future analyses of the effects of grafting on the content of secondary metabolites.

## 1. Introduction

Grape *(Vitis vinifera* L.) is one of the world’s most widely planted fruits [1], and ‘Cabernet Sauvignon’ (CS), a popular winemaking variety, is one of the most planted grape varieties in Xinjiang. Resveratrol is a natural polyphenol in plants, and it is present in grapes, pines, thuja, cassia, peanuts, and other plants; the content of resveratrol is highest in grapes (as high as 0.36% of the dry weight) [2]. The content of resveratrol varies greatly among grape varieties, and the content of resveratrol varies among tissues and growth stages within varieties; the resveratrol content is higher in the fruit peduncle, leaf, and peel than in other tissues [3,4], and winemaking varieties contain more resveratrol than fresh varieties [5]. The content of resveratrol is higher in American grape berries and leaves than in European grapes. Resveratrol is also closely related to the disease resistance of grapes; Dercks et al. found that the resveratrol content is five times higher in grape varieties with downy mildew resistance than in downy mildew-sensitive grape varieties [6]. Pezet et al. found that the resveratrol content is higher in resistant grape varieties infected with disease-causing bacteria than in sensitive grape varieties [7]. Glycosylation is one of the most common modifications in plant secondary metabolites, and large amounts of resveratrol accumulate in grapes in the form of glycosylation. The hydroxyl group in the natural resveratrol molecule combines with glucose to form piceid. Piceid has the same biological activity as resveratrol, and piceid can be converted to resveratrol under the action of glycosidase. This glycosidation protects resveratrol from oxidation by polyphenol oxidase, increases resveratrol’s water solubility, maintains its activity in cells, and increases transport and accumulation capacity [8,9]. Piceid can be hydrolysed to the more active resveratrol when grapes are stressed [10].

Grafting technology has been widely used in viticulture production; grafting affects the growth and development of grapes, which is closely related to their yield and quality. Zhang et al. found that different rootstock varieties can increase the light energy utilization index of scion to different degrees, which is conducive to improving the photosynthesis of scion CS [11]. Li et al. found that rootstocks of CS grapes had improved growth potential, fruit quality, and net photosynthetic rate [12]. The content of resveratrol, an indicator of fruit quality, is also altered in rootstocks. Zhai et al. used eight grape rootstocks in combination with CS and found that the resveratrol content and C4H and 4CL activities of grape leaves were increased in the rootstocks [13]. He et al. used five rootstocks grafted with CS grapes and CS autogenous seedlings (CK) and found that resistant rootstocks could increase the resveratrol content of CS grapes and the activities of synthesis-related enzymes in the seed and pericarp [14].

In previous studies, the mechanisms by which rootstocks affect the scion include the horizontal transfer of genes, the movement of mRNAs and sRNA, and the signalling of phytohormones [15,16,17]. MicroRNAs (miRNAs) are a class of endogenous non-coding small molecule RNAs in eukaryotes with lengths of 18–24 nt; they are formed by the shearing of single-stranded RNA precursors with a hairpin structure [18,19]. miRNAs exert post-transcriptional regulatory effects by cleavage degradation or translational repression of their target mRNAs through base-pairing with their target mRNAs [20]. The grafting of fruit trees such as apple and citrus can alter the expression of miRNAs such as miR156, miR171, miR172, and their target genes in the scion and root system, which can alter the growth and development of the root system and apical meristematic tissue [21,22]. A large number of miRNAs have been documented in grapes, and they play a role in regulating grape growth and development [23]; miR171 can have regulatory effects on bud differentiation in grapes [24], and VqmiR171c and VqmiR171i and their target gene *VqMYB154* can regulate the accumulation of stilbenoid synthesis substances in grapevines [25]. Currently reported miRNA target genes include MYB, WRKY and SPL family transcription factors [26] and *TAS* genes [27], NLR genes [28], and others, and most of these transcription factors can regulate the accumulation of stilbene biosynthesis in grapevines [29,30]. miRNAs can be transported to other tissues to exert their effects through the phloem and intercellular connective filaments, and large quantities of miRNAs have been detected in the phloem juices of pumpkin, cucumber, cucurbits, and apples, suggesting that they can be transported over long distances to exert their signalling effects in the phloem [31,32]. Rootstocks have been shown to affect the content of resveratrol in grape leaves in previous studies, and few studies have examined whether grape miRNAs are involved in the rootstock-mediated regulation of resveratrol in the scion. Here, CS grafted on the resistant rootstock 140R, self-grafted grapevine of the resistant rootstock 140R, and self-grafted grapevine of CS were used to identify miRNAs responsive to the grafting process via miRNA sequencing. We then analysed the roles of miRNAs and their target genes in rootstocks in mediating changes in the resveratrol content of the scion.

## 2. Results

### 2.1. Aboveground Morphology of Grapevine Plants

Grafting plays an important role in regulating the reproductive growth of the scion. CS/140R plants were stronger and had more leaves 2 months after grafting compared with CS/CS plants; the internode spacing of CS/140R plants was also smaller than that of CS/CS plants. This suggests that the rootstock affected the growth and development of the scion (Figure 1).

### 2.2. Resveratrol and Piceid Content of Grape Leaves

The resveratrol content and piceid content in grape leaves were determined using HPLC. In 2022, the resveratrol content and piceid content of CS/140R leaves were significantly higher than those of CS, which were 3.98 μg/g and 11.13 μg/g. In 2023, the changes to the resveratrol and piceid content of CS/140R leaves were consistent with those in 2022. It was concluded that grafting could increase the resveratrol and piceid content in CS/140R leaves. (Figure 2).

### 2.3. Sequencing and Analysis of sRNA Libraries

To identify miRNAs that respond to the grafting process, a total of nine sRNA libraries of three combinations of CS/140R, 140R/140R, and CS/CS were constructed; the libraries were sequenced using the Illumina platform. Among the nine libraries tested, 29.4 M, 38.45 M, 36.1 M, 35.32 M, 38.55 M, 34.94 M, 37.24 M, 40.2 M, and 42.99 M raw reads were generated from CS/140R-1, CS/140R-2, CS/140R-3, 140R/140R-1, 140R/140R-2, 140R/140R-3, CS/CS-1, CS/CS-2, and CS/CS-3, respectively; after filtering for impurities and removing low-quality sequences and polyA sequences, a total of 20.64 M, 20.64 M, 27.56 M, 26.5 M, 20.27 M, 21.06 M, 22.23 M, 30.12 M, 32.51 M, and 34.82 M clean reads were obtained, respectively. The Q30 of each sample was ≥ 95%. All the clean reads were compared with the reference genome of grapevine, Vitis_vinifera.PN40024.v4.53, and the reads containing genes were compared with the gene sequences. The ratio of reads to clean reads was 66.45–75.73%, and the quality of the data met the requirements for subsequent analysis (Appendix A). Clean reads from the grape leaves of different test materials were compared against the Vitis_vinifera.PN40024.v4.53 genome database. Clean reads were annotated to five different categories. Some of the unannotated sRNAs might be new miRNAs in grapevine leaves (Appendix A).

The Dicer enzyme and DCL enzyme have a strong bias for the first base pair U at the 5′ end when recognising and cleaving precursor miRNAs. Typical miRNA base ratios were obtained by base preference analysis of miRNAs. The 5′ end first base preference and the base preference of each locus of the known miRNAs, newly predicted miRNAs, and overall miRNAs are shown in Figure 3. The first base preference of miRNAs was stronger than that for U, which is in accordance with the general base preference of miRNAs, indicating that the miRNAs obtained by sequencing are of high quality and high confidence (Figure 3).

### 2.4. Identification of Grapevine miRNAs

A total of 177 known miRNAs, belonging to 46 miRNA families, were identified through comparison with the miRBase18.0 (http://www.mirbase.org) database (Appendix A). The lengths of the sRNAs of known miRNAs were predominantly at 21 nt, and the lengths of the sRNAs of Novel miRNAs were predominantly at 21 nt and 24 nt (Figure 4A,B). In addition, most of the known miRNA families had multiple members, such as vvi-miR169 with twenty-five members; vvi-miR395 with fourteen members; vvi-miR156 with nine members; vvi-miR171 with nine members; vvi-miR845 with three members; and vvi-miR162 with only one member (Figure 4C).

The miRDeep2 software package (mirdeep2 2.0.1.3) was used to obtain possible precursor sequences based on the position information of the reads determined from the genome comparison, the distribution of the reads on the precursor sequences (based on the characteristics of the miRNAs, including their star and loop structures), and structural energy information. Novel miRNAs were then predicted using the Bayesian model. A total of 267 novel miRNAs were identified from grape leaves of different test materials (Appendix A). To identify novel miRNAs, sRNA sequences were localised to the reference genome of grape. Using perfectly matched sRNA sequences, stem-loop structure prediction was performed using the M-folding procedure. The results showed that 89 sRNA sequences perfectly matched and folded into stem-loop structures; the structures of some novel miRNA precursors are listed in Figure 4D.

### 2.5. Quantitative Analysis of Grapevine miRNAs

The expression of miRNAs in each sample was determined and normalised using the TPM algorithm. The distribution of miRNA expression describes the overall miRNA expression pattern in each sample. In each sample, the log10TPM of miRNAs was predominantly between 1 and 3 (Figure 5A). The miRNA counts per sample ranged from 373 to 416 (Figure 5B). The expression levels of miRNAs in the three replicates of the samples roughly converged, indicating that the biological reproducibility was high.

### 2.6. Differential miRNA Analysis

Differentially expressed miRNAs between treatments were identified using the following criteria: FC ≥ 2 and FDR < 0.05; FC indicates the ratio of expression between two samples (groups). The results of identifying differential miRNAs are shown in Appendix A. By analysing the differential expression of miRNAs in grapevine leaves, it was found that there were 125 differential miRNAs in the leaves of CS/140R rootstock combination compared with CS/CS, among which 57 differential miRNAs were up-regulated, such as the expression of vvi-miR3624-3p, novel_miR_181, and vvi-miR3627-5p which were significantly up-regulated, and 68 differential miRNAs were down-regulated, such as vvi-miR171c, novel_miR171c, and novel_miR3627-5p. There were 68 down-regulated differential miRNAs, such as vvi-miR171c, novel_miR_166, novel_miR_128, vvi-miR169c, and vvi-miR828a. Compared with CS/CS, there were a total of 276 differential miRNAs in 140R/140R leaves, among which 114 up-regulated differential miRNAs, such as novel_miR_1, vvi-miR398a, vvi-miR479, and vvi-miR319f. There were 162 down-regulated differential miRNAs, such as novel_miR_110 and novel_miR_175. Many miRNAs responsive to the grafting process, such as VvmiR169c, VvmiR171c, and VvmiR828a were significantly down-regulated, suggesting that these miRNAs may be affected by grafting, indicating that miRNAs play a complex role in regulating scion growth after grafting (Table 1).

### 2.7. miRNA Target Gene Analysis and Functional Annotations of the Differentially Expressed miRNAs

To further clarify the biological regulatory functions of miRNAs and target genes in grapevine leaves, miRNA target genes were predicted based on known miRNA sequences, new miRNA sequences, and gene sequences of corresponding species. The results showed that the 444 miRNAs predicted 5437 target genes. Annotation information of target genes was obtained using BLAST software (blast+2.14.0), and 5373 target genes out of 5437 target genes were annotated in the different databases (Appendix A). The top three COG functional classes were Carbohydrate transport and metabolism (179), General function prediction only (216), and Signal transduction mechanisms (189). The top three KOG functional classes were Carbohydrate transport and metabolism (130), General function prediction only (585), and Signal transduction mechanisms (390) (Appendix A).

In the GO enrichment analysis of differentially expressed miRNAs, GO terms in the cellular component category included Membrane, Membrane fraction, Cell, and Cell fraction; GO terms in the molecular function category included binding, catalytic activity, transporter activity, and nucleic acid-binding transcription factor activity; GO terms in the biological process category included cellular metabolic processes and metabolic process (Figure 6B). KEGG analysis revealed 489 differentially expressed target genes involved in metabolic pathways (26), genetic information process pathway (14), tissue system pathway (2), environmental information process pathway (4), and cellular process pathway (3) (Figure 6A). Eight genes were enriched in the phenylalanine metabolism pathway (Phenylalanine metabolism). Many of these target genes encode stress-related transcription factors, such as growth hormone response factor (ARF) family members, nuclear transcription factor Y, and MYB transcription factors. Target genes involved in other metabolic pathways, including TMV resistance proteins and ribosomal RNA processing proteins, were also identified.

### 2.8. RT-qPCR Verification of Differentially Expressed miRNAs

Based on differential miRNA analysis, eight known miRNAs and one newly identified miRNA with significant expression induced by grafting were selected for expression analysis. The results showed that the qRT-PCR results were highly consistent with the trend of miRNA-Seq results, confirming the reliability of miRNA-Seq results (Figure 7). Compared with CS/CS, vvi-miRNA171c, vvi-miRNA171i, vvi-miRNA171d, vvi-miRNA398a, vvi-miRNA828a, vvi-miRNA3639-5p, and novel-miR-93 were differentially down-regulated in CS/140R. The expression of vi-miRNA156b and vi-miR156i was up-regulated; in 140R/140R, the expression of vi-miRNA171c, vi-miRNA171i, vi-miRNA398a, and vi-miRNA3639-5p was significantly down-regulated and to a lesser extent than that of CS/140R. It was hypothesized that 140R rootstock may regulate scion growth, development, and metabolism by affecting miRNA expression.

### 2.9. Functional Characterisation of vvi-miR171c

Target gene prediction of vvi-miR171c showed that vvi-miR171c was paired with *VvMYB154* (Vitvi11g00228) (Figure 8A), which has been reported in the literature to regulate resveratrol synthesis [25]. The relative expression levels of vvi-miR171c and its target gene *VvMYB154* were detected by RT-qPCR. The relative expression of vvi-miR171c was significantly up-regulated, and that of *VvMYB154* was significantly down-regulated in pCa-vvi-miR171c-transformed leaves compared with the empty vector control (CK) and CS (Figure 8A). The resveratrol content of CS, CK, and pCa-vvi-miR171c-treated leaves was determined by HPLC, and the results are shown in Figure 8B. The resveratrol content and piceid content were significantly lower in pCa-vvi-miR171c-treated leaves than in CS-treated and CK-treated leaves (1.04 µg/g and 2.01 µg/g).

## 3. Discussion

Grafting, an ancient agricultural technique, is widely used in the cultivation and production of grapes, and the selection of suitable rootstocks for grafting can improve plant growth rates, fruit quality, and resistance to abiotic stress. In this study, the content of resveratrol and piceid in the leaves of CS self-grafted grapevines, self-grafted grapevines of the resistant rootstock 140R, and CS/140R grafted grapevines were determined, and the expression of miRNAs under grafting conditions was investigated. The results showed that the content of resveratrol and piceid was significantly higher in the leaves of CS/140R than in CS self-grafted grapevines. This suggests that the rootstock 140R can alter the content of resveratrol in the leaves of the scion through grafting, which is consistent with the results of a previous study [13].

miRNAs are endogenous, non-coding small RNAs that regulate plant growth, development, and morphogenesis [33]; they also play a role in responses to various abiotic and biotic stresses by regulating the expression of target genes at the post-transcriptional level [34,35]. Mutual grafting has been shown to alter the abundance of several miRNAs and predicted target genes in Fuji and M9 apples [36]. Given that grape miRNAs respond to grafting and are differentially expressed in specific genotypes, mutual grafting of these genotypes can alter the abundance of these miRNAs in the scion or rootstock to improve scion tolerance and quality [37,38]. In our study, miRNA sequencing of grafted grapevines revealed significant differences in the abundance of miRNAs in the leaves of the scion of CS and CS self-grafted grapevines. This indicates that grafting can alter the abundance of miRNAs. A total of 177 known miRNAs and 267 new miRNAs were identified in this study, and the sequences of the obtained miRNAs were consistent with the secondary structure criteria of miRNAs [39,40]. The miRNAs in the nine libraries were all greater than 21 nt in length. miRNAs 21 nt and 24 nt in length were the most abundantly expressed in the libraries, and they might play an important role in mediating the effects of grafting on the growth and development of the scion. A more in-depth study of miRNA transcriptomes revealed that the size distributions of the miRNAs were similar to those observed in other plants, such as apple [41] and pomegranate [42], which had the highest proportion of 21–24 nt miRNAs. In addition, we found that 21 nt miRNAs were the most abundant, not 24 nt miRNAs, which is consistent with the results of a previous study in which grape miRNAs were sequenced [43]. These results suggest that the miRNA transcriptome is similar and specific across plant species and that miRNAs with different lengths have different regulatory effects on the expression of genes; furthermore, these findings suggest that the regulatory effects of miRNAs on gene expression during the response to grafting are broad, and this can have implications for the scion.

Many widely distributed and highly conserved miRNAs have been identified in plants, and some of these miRNAs occur only in specific plant species [40]. Previous studies have shown that the diversity of miRNA families is determined by the abundance and number of family members [36,44]. In our study, 177 miRNAs are known to belong to 46 conserved miRNA families, and these miRNAs are widely distributed in dicots and monocots [45,46]. These known miRNAs have more than one family member, such as miR169 (25 members), miR395 (14 members), miR156 (9 members), miR171 (9 members), miR159 (3 members), miR166 (8 members), and miR634 (4 members). Wei et al. [47] similarly found that a total of 3027 miRNAs were identified in grapes under salt stress conditions. Of these, 174 miRNAs were found to be highly conserved. A previous study has shown that the expression of miRNAs in the scion is altered significantly after apple grafting [36]. According to the KEGG analysis, the target genes of differentially expressed miRNAs were enriched in the phenylpropanoid synthesis pathway in the CS/CS vs. CS/140R and CS/CS vs. 140R/140R comparison groups. To confirm the reliability of the sequencing results, we used RT-qPCR to detect the expression of some representative genes associated with the response to grafting. These findings suggest that grafting affects the expression of miRNAs in grapevines, which in turn affects the synthesis of substances in the scion.

miRNAs have been found to play key regulatory roles in plant growth and development at the post-transcriptional level. miR171 family members were the first miRNAs to be discovered, and they regulate the responses of chloroplasts to salt stress [48]. Ma et al. [49] found that *SCL27*, a target gene of miR171, in *Arabidopsis thaliana* interacts with the cis-element of the chlorophyll oxidoreductase (PORC) promoter and is involved in chlorophyll synthesis. miR171c responds to exogenous GA3 application under salt stress, which regulates the development of grape seeds [50]. Jiang [25] found that the expression of *MYB154* is negatively regulated by vqmiR171c and vqmiR171i in the Chinese wild hairy grape ‘Danfeng-2’, which affects the accumulation of astragalus, and pathogenic bacteria can inhibit the expression of vqmiR171c and vqmiR171i, which increases the stilbenoid synthesis content. In our study, the expression of vvi-miR171c was higher in 140R than in CS, and the expression of its target gene, *VvMYB154*, exhibited the opposite pattern. The vvi-miR171c regulatory module was hypothesised to negatively regulate the increase in the resveratrol content in grapes, which suggests that the grafting treatment affected the expression of vvi-miR171c and increased the resveratrol content. We also generated transiently transformed CS grape leaves. Overexpression of vvi-miR171c in grape leaves altered the expression of the target gene *VvMYB154*. This suggests that vvi-miR171c regulates the expression of *VvMYB154* under grafting conditions to increase the resveratrol content.

## 4. Materials and Methods

### 4.1. Experimental Site and Material Culture

In October 2021, CS and 140R (*V. berlandieri* × *V. rupestris*) branches were collected and placed in plastic bags; they were then sealed and stored in low-temperature sand; in January 2022, the CS and 140R branches that had been stored in low-temperature sand were cut into 10–15 cm sections (including one bud) for hard grafting. The grafted CS/CS, 140R/140R, and CS/140R branches were placed in a hotbed to induce rooting and germination; after 20 days, the branches were placed in 8 cm × 8 cm × 10 cm nutrient pots filled with seedling substrate (peat:vermiculite = 2:1, *v*:*v*). They were then placed in a greenhouse and subjected to standard management conditions (light intensity of 12,000 Lx, 16 h/8 h light/dark photoperiod, relative humidity between 85% and 90%; and temperature of 28 °C (day)/18 °C (night)). Three replicates of each treatment were performed, and there were 10 grapevines per replicate. After two months, plants with uniform growth (10–11 leaves in the scion) were selected for sampling, and the 4th to 6th functional leaves from the bottom of the scion were collected and washed with deionised water; they were then dried, frozen in liquid nitrogen, and stored in an ultra-low-temperature refrigerator at −80 °C for small RNA sequencing and leaf resveratrol content determination. To account for possible experimental error, the previous year’s experiment was repeated in October 2022 until the scion had 10–11 leaves, and the resveratrol content and piceid content were determined.

### 4.2. Determination of the Res and Piceid Content

The composition of the resveratrol standard solution was based on that described by He Wang [51]; the standard curve is shown in Table 2. The content of resveratrol was determined using a Shimadzu high-performance liquid chromatograph (LC-2010AHT). One g of the sample was ground in liquid nitrogen into powder, transferred to a centrifuge tube, fixed in methanol to a volume of 10 mL, vortexed for 2 min, ultrasonicated for 30 min, and then centrifuged at 12,000 rpm for 10 min; this process was then repeated, and the supernatant was filtered through a 0.45 µm membrane (organic phase) in a vial (Agilent). It was then placed in a refrigerator at 4 °C. The mobile phase was acetonitrile/0.2% phosphoric acid = 45/55, the detection wavelength was 306 nm, the flow rate was 0.8 mL/min, the column temperature was 25 °C, the injection volume was 10 μL, and the elution velocity was low.

### 4.3. Small RNA Library Construction and Sequencing Analysis

#### 4.3.1. RNA Extraction and sRNA Library Construction

RNA was extracted from grape leaves and the quality and concentration of RNA in grape leaves were determined using BioMike Biotechnology Co., Ltd. (BioMike, Beijing, China). Nine small RNA (sRNA) libraries were constructed following the method of Zhu et al. [52]; specifically, the sRNAs were isolated from the total RNA using a PAGE gel, and 18–30 nt bands were recovered; sRNAs were then ligated to the 5’ and 3’ junctions and reverse-transcribed to synthesise sRNAs, followed by sequencing. The sRNAs were ligated to the 5’ and 3’ junctions and reverse-transcribed to synthesise cDNA; following amplification via PCR and separation from the target DNA fragments by PAGE gel electrophoresis, they were used to construct the cDNA library. Sequencing was performed on an Illumina platform.

#### 4.3.2. sRNA Annotation and miRNA Identification

The raw data obtained from sequencing (Raw reads) were subjected to quality control and de-jointed; sequences shorter than 18 nt or longer than 30 nt were removed, sequences with low-quality values were removed from each sample, and reads with an unknown base N (N is an unrecognisable base) content greater than or equal to 10% were removed to obtain high-quality sequences (Clean reads). Using Bowtie [53] software (Bowtie 2.3.5.1), Clean Reads were compared against the Silva database, GtRNAdb database, Rfam database, and Repbase database, and ribosomal RNAs (rRNAs), transfer RNAs (tRNAs), small intranuclear RNAs (snRNAs), small nucleolar RNAs (snoRNAs), ncRNAs, and repetitive sequences were removed to obtain Unannotated reads containing miRNAs. Vitis_vinifera.PN40024.v4.53 [54] was used as the reference genome for sequence comparison and subsequent analysis. The Unannotated reads were aligned to the reference genome using Bowtie software to obtain position information; these were referred to as Mapped Reads.

To identify known miRNAs, the reads matched to the reference genome were compared with the mature sequences of known miRNAs and their upstream 2 nt and downstream 5 nt in the miRBase (v22) database, with a maximum of one mismatch permitted.

MiRNA transcription start sites are more frequently found in intergenic regions, introns, and reverse strand of coding regions. miRNA genes are firstly transcribed into primary miRNA (pri-miRNA) and processed into precursor miRNA (pre-miRNA), which is characterized by its hair-pin structure, and finally matured into miRNA with help from the Dicer/DCL enzyme. In miRDeep2 [55] modules, potential miRNA precursors were extracted from reference genome based on reads mapping. A further selection of potential precursors counts on RNA secondary structure, where candidate precursors are expected to be able to be partitioned into candidate mature, loop, and star part based on reads mapping. RNAfold randfold *p*-value will be given to a subset of potential precursors. Each precursor will be scored by Bayesian statistics to describe the fit of reads to the biological model of miRNA biogenesis. This software is mainly designed for animal miRNA prediction; however, by adjusting the parameters and algorithm, plant miRNA prediction can also be achieved. Based on the sequence similarity, analysis of miRNA families was performed on the detected known miRNAs and new miRNAs to study the extent to which the miRNAs are conserved [56].

#### 4.3.3. Analysis of Differentially Expressed miRNAs

Differentially expressed miRNAs were detected using the following criteria: |log2 (fold change (FC))| ≥ 1.00 and false discovery rate (FDR) ≤ 0.05. FC indicates the ratio of expression between two samples (groups) [57,58].

#### 4.3.4. miRNA Target Gene Prediction and Functional Annotation

Based on the gene sequence information of known miRNAs and newly predicted miRNAs in corresponding species, target genes were predicted using TargetFinder software (TargetFinder 1.1) [59]. The predicted target gene sequences were compared with the NR [60], GO [61], COG [62], KEGG [63], and KOG [64] databases to obtain annotation information of the target genes using BLAST software (blast+2.14.0) and identify genes with specific biological functions.

#### 4.3.5. RT-qPCR Verification of Differentially Expressed miRNAs and Their Target Genes

Total RNA was extracted from CS/CS, CS/140R, and 140R/140R-treated leaves according to the standard Trizol protocol (Invitrogen, Carlsbad, CA, USA). gDNA was removed and reverse-transcribed using the PrimeScript™ RT reagent Kit (Takara, Kusatsu, Japan) per the manufacturer’s instructions. Gene-specific primers for qRT-PCR were designed using Vector NTI 10 software, and the relative expression of genes was set to 1 using CS/CS leaves as the reference sample. Relative expression was calculated using the 2^−ΔΔCt^ method [65], and U6 was used as an internal standard to normalise the expression of miRNAs. Eight known miRNAs and one novel miRNA were selected to verify the expression of miRNAs inferred using RNA-seq. Primers used for miRNAs in qRT-PCR experiments are shown in Appendix A.

#### 4.3.6. Plasmid Construction and Generation of Transgenic Plants

The precursor sequence of vvi-miR171c was amplified and cloned into the pCambia1301 vector containing the CaMV35S promoter. This was confirmed by sequencing prior to introduction into Agrobacterium tumefaciens GV3101; the primers used are shown in Appendix A.

#### 4.3.7. Transient Transformation of vvi-miR171c in Grapevine

Annual cuttings of CS grapevines were used. Leaves of plants with a height of 15 cm were used for vvi-miR171c overexpression. Each treatment consisted of three biological replicates with three plants per biological replicate. Suspended Agrobacterium solutions were soaked in grapevine leaves (−0.1 Mpa, 30 min in vacuo). They were incubated for 2 days, washed with deionised water and dried, snap-frozen in liquid nitrogen, and stored in an ultra-low temperature refrigerator at −80 °C [29,43].

#### 4.3.8. Statistical Analyses

Statistical analyses of the data, including multiple comparisons, were conducted using Microsoft Excel 2010 (Microsoft Corporation, Redmond, WA, USA) and SPSS 19.0 (IBM). Data in the graphs and charts of Origin 2018 software (OriginLab Inc., Northampton, MA, USA) were presented as mean ± standard deviation (Mean ± SD). More than three replications were performed for each indicator.

## 5. Conclusions

A total of 177 known miRNAs and 267 novel miRNAs were identified in this study. Many miRNAs responsive to the grafting process, such as vvi-miR169c, vvi-miR171c, vvi-miR171e, and vvi-miR828a, were significantly down-regulated in CS/140R leaves compared with CS/CS leaves, suggesting that the expression of these miRNAs might be affected by grafting. KEGG analysis revealed that miRNAs regulated the expression of genes in the phenylalanine metabolic pathway; the expression of vvi-miR171c was down-regulated and the expression of *VvMYB154* was up-regulated in CS/140R leaves. Grapevine leaves transiently overexpressing vvi-miR171c were further assayed; the expression of the target gene *VvMYB154* was down-regulated, and the resveratrol content was decreased, indicating that vi-miR171c regulates the expression of *VvMYB154* under grafting conditions and affects the resveratrol content in grapes. In sum, 140R regulates the expression of vvi-miR171c under grafting conditions to increase the resveratrol content of the scion. These results provide new information that will aid future studies of how grafting affects the content of secondary metabolites in the scion and scion growth and development. (See Figure 9).

## Figures and Tables

**Figure 1 plants-13-03057-f001:**
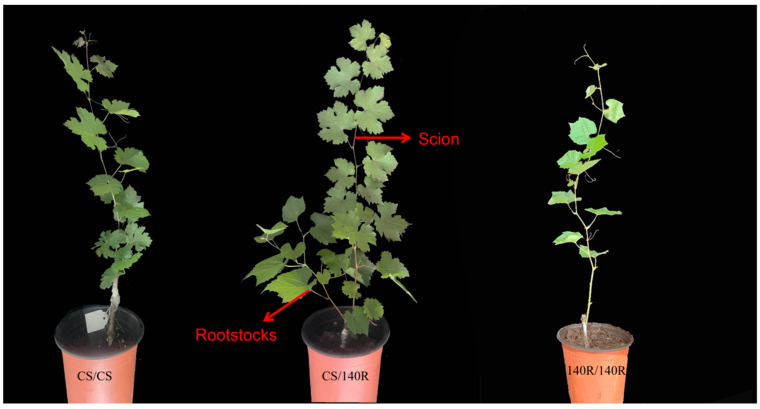
Test material: Grape plant growth morphology (photographed 60 days after grafting, scale bar is 10 cm).

**Figure 2 plants-13-03057-f002:**
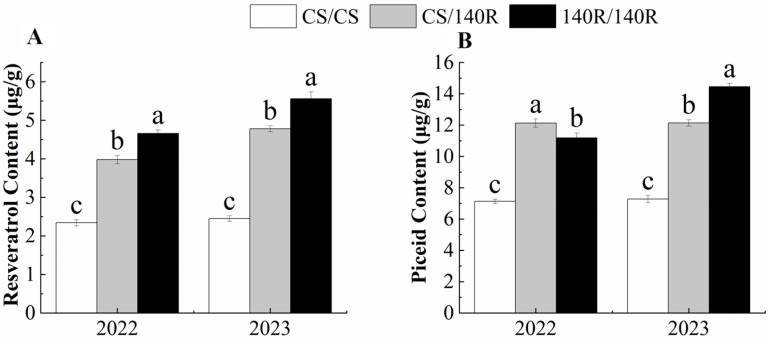
Resveratrol and piceid content. (**A**): resveratrol content; (**B**): piceid content. Note: Letters show difference between treatments by the 5% Tukey’s test. Values represent means ± standard deviation of three replicates, error bars represent SD (*n* = 3). the proportion of resveratrol in the dry weight.

**Figure 3 plants-13-03057-f003:**
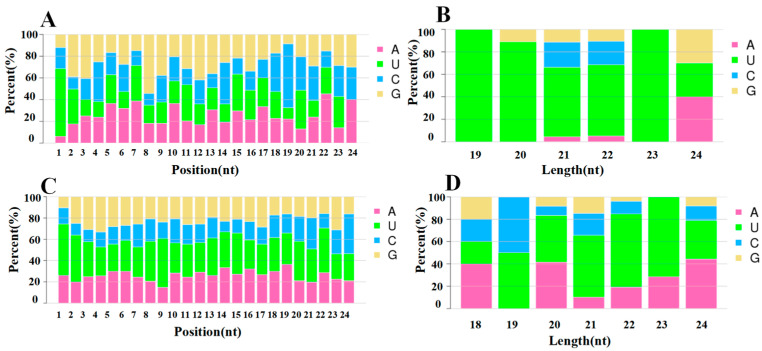
Analysis of the miRNA nucleotide bias. (**A**,**C**) Base preference on known and novel miRNA at each position, respectively. (**B**,**D**) First base preference of known and novel miRNA in different length, respectively.

**Figure 4 plants-13-03057-f004:**
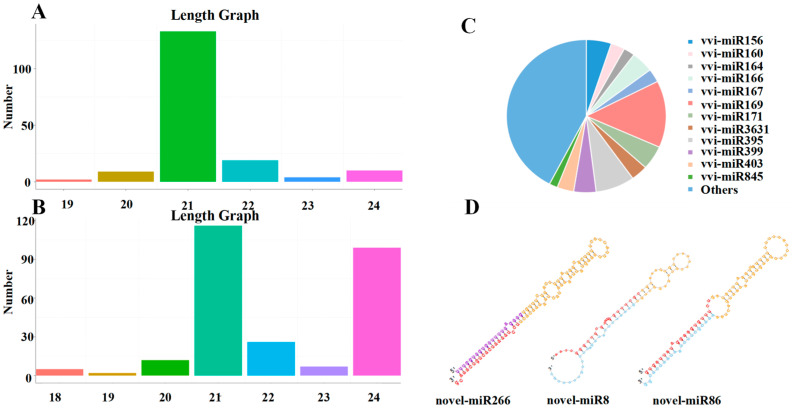
The length distribution and family of miRNAs identified in grapes. (**A**,**B**) Length distribution of known and novel miRNAs. (**C**) Analysis of the miRNA family. (**D**) Secondary structures of some novel miRNAs (Novel miR266, Novel miR8, and Novel miR86). Note: The secondary structure contains the position of mature sequence, ring structure, star sequence, purple is the star sequence predicted by the miRDeep2 software, bright blue is the star sequence supported by sequencing reads.

**Figure 5 plants-13-03057-f005:**
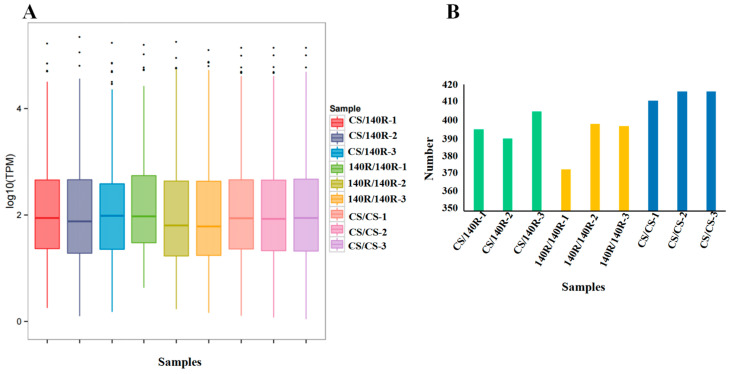
Analysis of miRNA expression status in each sample. (**A**) Boxplot of the overall distribution of miRNA expression in each sample. (**B**) miRNA statistics in each sample.

**Figure 6 plants-13-03057-f006:**
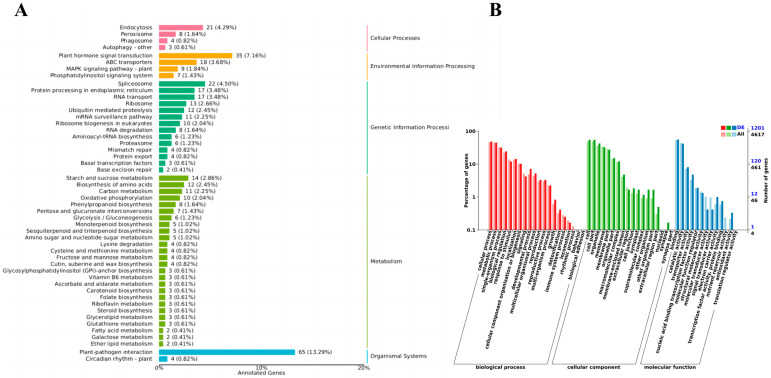
Annotation of miRNA target genes (**A**) differential miRNA KEGG analysis (**B**) differential miRNA GO analysis.

**Figure 7 plants-13-03057-f007:**
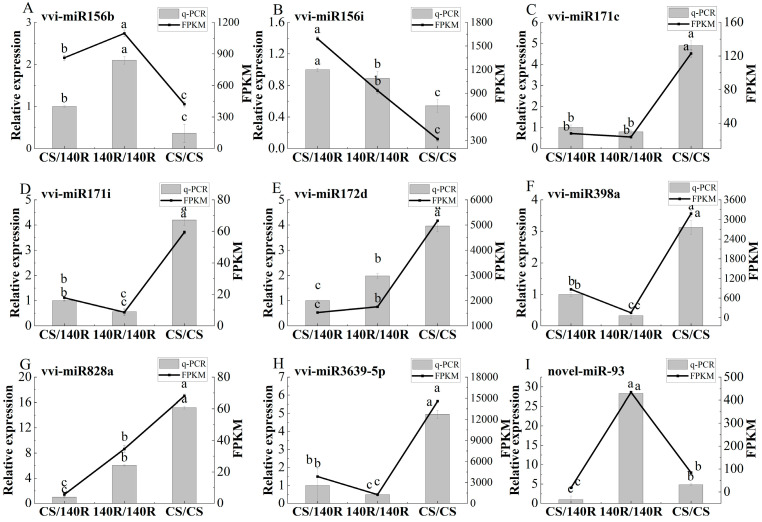
Quantitative real-time PCR (qRT-PCR) and sequencing (miRNA-Seq) results of nine candidate miRNAs. (**A**): vvi-miR156b; (**B**): vvi-miR156i; (**C**): vvi-miR171c; (**D**): vvi-miR171i; (**E**): vvi-miR172d; (**F**): vvi-miR398a; (**G**): vvi-miR828a; (**H**): vvi-miR3639-5p; (**I**): novel-miR-93. Letters show difference between treatments by the 5% Tukey’s test. Values represent means ± standard deviation of three replicates, error bars represent SD (*n* = 3).

**Figure 8 plants-13-03057-f008:**
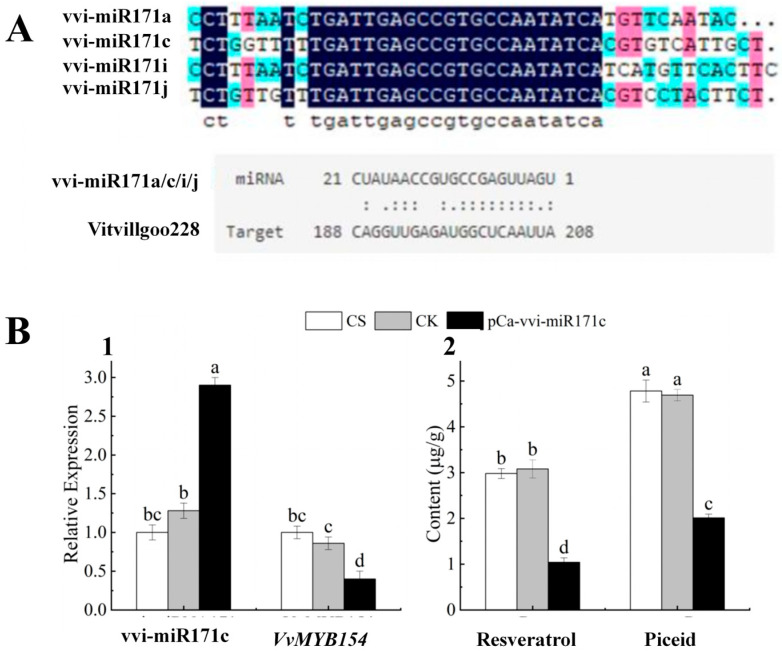
Expression analysis of vvi-miR171c (**A**): miRNA precursor sequence alignment analysis and prediction of target gene interaction list (**B**): vvi-miR171c negatively regulates the expression of *VvMYB154* in grapevine (**1**) RT-qPCR was used to detect the relative expression levels of vvi-miR171c and its target gene *VvMYB154*. CS: normal grapevine; CK: empty carrier; pCa-vvi-miR171c: overexpression of vvi-miR171c. (**2**) Resveratrol content and piceid content in CS, CK, and pCa-vvi-miR171c). Note: Letters show difference between treatments by the 5% Tukey’s test. Values represent means ± standard deviation of three replicates, error bars represent SD (*n* = 3).

**Figure 9 plants-13-03057-f009:**
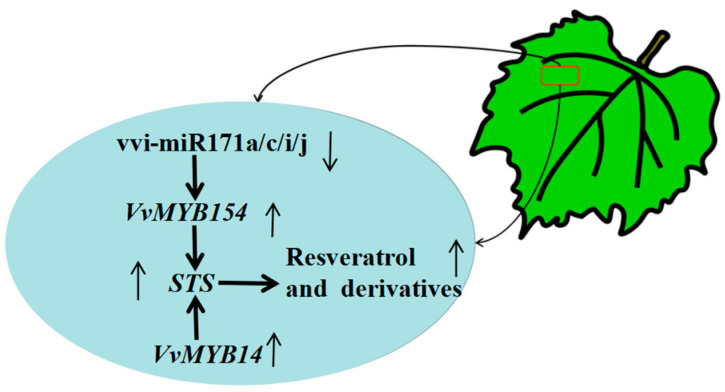
Schematic diagram of CS/140R Cabernet Sauvignon leaf miRNA and target genes.

**Table 1 plants-13-03057-t001:** Differentially expressed miRNAs in grape leaves of each comparison group.

miRNA Name	log2(CS/CS vs. CS/140R)	miRNA Name	log2FC(CS/CS vs. 140R/140R)
vvi-miR171c	−2.055713777	novel_miR_110	−3.638579789
vvi-miR169c	−3.99381221	novel_miR_175	−2.462009742
novel_miR_166	−3.908111265	vvi-miR828a	−1.703482327
vvi-miR828a	−3.383976384	novel_miR_177	−1.702342234
novel_miR_128	−3.043316763	vvi-miR169l	−1.555720976
novel_miR_2	−3.043316763	novel_miR_1	9.725505306
novel_miR_263	2.734070564	vvi-miR398a	4.280026387
novel_miR_172	2.86081601	novel_miR_183	3.475394554
novel_miR_181	2.86081601	vvi-miR479	2.744971769
vvi-miR3627-5p	3.168791739	vvi-miR319c	2.024953876
vvi-miR3627-3p	3.412211104	novel_miR_52	2.03448677
vvi-miR3624-5p	3.465130458	vvi-miR319f	2.036424038
vvi-miR3624-3p	3.545753663	vvi-miR399e	2.199961988

**Table 2 plants-13-03057-t002:** Retention time and regression equation of trans-Res, trans-Pd.

Compound	Retention Time	Standard Curves	Coefficient of Determination
trans-Res	2.931	y = 132,447x − 120,557	0.9993
Trans-Pd	4.328	y = 103,081x − 74,534	0.9996

## Data Availability

Data are contained within the article.

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
