# Peer review of "‘140R’ Rootstock Regulates Resveratrol Content in ‘Cabernet Sauvignon’ Grapevine Leaves Through miRNA"

_plants, 2024, doi:10.3390/plants13213057_

Round 1
Reviewer 1 Report
Comments and Suggestions for Authors
The authors Zhu et al. evaluated the influence of rootstocks for grafting in Cabernet Sauvignon grape and microRNA’ level changes in leaves after such experiments. The topic of this research is actual and the manuscript may be interesting for specialists in this field. The introduction provides a solid background for the results presented in the paper, and the cited literature is appropriately selected and relevant to the study.
I raise the following comments to the manuscript:
Figure 2 B. Please check the statistical difference and letters labelled above the bars in this Figure (2B, left graph). I guess they are labelled not according to the statistical analysis. The information about the type of statistical analysis can be included in the figure legend.
Line 37, line 106. Please avoid using the “Res” instead of Resveratrol here and in some other sentences the text.
Lina 81-82. Please modify the phrase “Most of the miRNA target genes reported to date are transcription factors”. A lot of others targets were found for miRNA such as TAS genes, NLR genes and etc.
Authors should unify the names of miRNA in the text. Sometimes, they used the VqmiR171 or others, but sometimes - vqmiR171c or others.
Line 117. Table 1. The abbreviations such as scRNA, snRNA, snoRNA, and others should be clearly deciphered in the table legend.
Line 150/ Please change the word “known”.
Figure 8. The information about statistical analysis as well as the number of probes is absent. Please show bars in the figure and present the additional information in the figure legend.
Line 315. Please use italics for Latin names.
Author Response
Thank you very much for your scientific rigor in reviewing the manuscript.
The following are responses to the comments:
Comments 1: Figure 2 B. Please check the statistical difference and letters labelled above the bars in this Figure (2B, left graph). I guess they are labelled not according to the statistical analysis. The information about the type of statistical analysis can be included in the figure legend.
Response 1:Thank you for pointing this out. I've modified this section to add methods of data processing.
Comments 2: Line 37, line 106. Please avoid using the “Res” instead of Resveratrol here and in some other sentences the text.
Response 2:Thank you for pointing this out. I have completed the modifications.
Comments 3: Lina 81-82. Please modify the phrase “Most of the miRNA target genes reported to date are transcription factors”. A lot of others targets were found for miRNA such as TAS genes, NLR genes and etc.
Response 3:Thank you for pointing this out. I have revised this part of the content and added new references.
Comments 4: Authors should unify the names of miRNA in the text. Sometimes, they used the VqmiR171 or others, but sometimes - vqmiR171c or others.
Response 4:Thank you for pointing this out. I have completed the modifications. miRNA171 refers to a family. VqmiR171c and VqmiR171i appearing in line 79 of the article are named in the cited references, and the c and i following indicate that the two mirnas are highly homologous. In this study, according to the naming rules of miRNA, the mature miRNA is abbreviated as miR, and then represented as vvi-miRNA171c according to its species name plus the miRNA family studied, and the target gene is uniformly represented as VvMYB54. Based on the above, we have modified and confirmed the names appearing in the article.
Comments 5: Line 117. Table 1. The abbreviations such as scRNA, snRNA, snoRNA, and others should be clearly deciphered in the table legend.
Response 5:Thank you for pointing this out. In order to better present our results, we have streamlined the data from Table 1 and placed it in Supplementary Data Table 1 with annotations. Also this is explained in detail in 4.3.2 of the article (lines 397-400).
Comments 6: Line 150/ Please change the word “known”.
Response 6:Thank you for pointing this out. I have completed the modifications (line152).
Comments 7: Figure 8. The information about statistical analysis as well as the number of probes is absent. Please show bars in the figure and present the additional information in the figure legend.
Response 7:Thank you for pointing this out. I have completed the modifications.
Comments 8: Line 315. Please use italics for Latin names.
Response 8:Thank you for pointing this out. I have completed the modifications (line 329).
Reviewer 2 Report
Comments and Suggestions for Authors
Article: '140R' rootstock regulates resveratrol content in 'Cabernet Sauvignon' grapevine leaves through miRNA
This work is devoted to the study of microRNA in Cabernet Sauvignon grapevine plants grafted onto the resistant 140R rootstock, self-grafted vines of the resistant 140R rootstock, and self-grafted CS grapevines using high-throughput sequencing methods. It is shown that grafting has a positive effect on plant growth and development and significantly affects the miRNA loss. Overall, the work is interesting, but it needs to be significantly improved for publication.
Line 109. Are you indicating fresh or dry weight (FW or DW)? This must be indicated in the text and in the figure after the unit of measurement.
Figure 2. It is necessary to make the letter designations larger. In the figure caption, indicate the method of statistical processing.
I am also wondering if you analyzed the trans or cis form of resveratrol and piceid? Why didn't you analyze other stilbenes in grape leaves and roots? They usually contain viniferines and piceatannol.
How many plants were analyzed for stilbenes? Usually, stilbenes can vary greatly among grape plants, so it is difficult to get a small SD, this is normal. In this case, I see a very small SD, this raises questions. It looks like several technical replicates from the same plant. I would like to clarify this issue.
Figure 3. It is necessary to make the figures larger, since the legends are impossible to distinguish and it is difficult to evaluate the data. This remark applies to all figures in this article. Previously, in Figure 2, the authors used capital letters to denote figures, in this and subsequent figures, small letters are presented.
I do not understand why you provide data on scRNA and cnRNA in Table 1. To show that they do not exist?
2.6. Differential miRNA analysis. This should be very interesting data, but in fact you don't have it. In the text you describe this data poorly, and in the figure it is impossible to understand anything. This needs to be corrected.
2.8 RT-qPCR data. There is no statistical processing of the data at all. This is unacceptable. It is also necessary to describe the obtained results in detail. It is also necessary to use several housekeeping genes to normalize the data, this is the only way to reliably estimate the expression level.
Overall, the results are described poorly and insufficiently in detail, the discussion part of the work needs to be improved. The conclusions are poorly supported by the results.
Author Response
Thank you very much for your scientific rigor in reviewing the manuscript.
The following are responses to the comments:
Comments 1: Line 109. Are you indicating fresh or dry weight (FW or DW)? This must be indicated in the text and in the figure after the unit of measurement.
Response 1:Thank you for pointing this out. When resveratrol is measured, it needs to be freeze-dried, so we refer to dry weight, which is indicated in the figure notes.
Comments 2: Figure 2. It is necessary to make the letter designations larger. In the figure caption, indicate the method of statistical processing.
I am also wondering if you analyzed the trans or cis form of resveratrol and piceid? Why didn't you analyze other stilbenes in grape leaves and roots? They usually contain viniferines and piceatannol.
Response 2: TThank you for pointing this out, we have modified Figure 2. The font size of the letters has also been made larger.
Since trans-resveratrol is the form of resveratrol in its natural state and cis-resveratrol is the isomerized form produced by plants under the influence of ultraviolet light, but the cis-isomer is more photosensitive and highly decomposable, so it is difficult to be extracted and purified, so the resveratrol that we determined was mainly trans-resveratrol.
All resveratrol-related substances were measured in our previous study, and the data have not been published. The conclusion is consistent with the results of this experimental study, which found that the resveratrol content of Cabernet Sauvignon grapes grafted onto 140R rootstocks, Cabernet Sauvignon fruits, seeds, and leaves was significantly higher than that of ungrafted Cabernet Sauvignon. Before selecting grape leaves as the test material, we determined the content of resveratrol and resveratrol glycosides in grape leaves, fruit skins and seeds of 3-year-old CS/140R grafted seedlings, CS/CS grafted seedlings and 140R/140R grafted seedlings. The results revealed that the resveratrol content of the three treatments, leaf>seed>skin, which was consistent with the conclusion of He Wang (2020). In combination with the selection of leaves for miRNA sequencing and transient transfection of leaves to validate the function of the miRNA171c gene, the leaves were selected as the test material.
When we measured resveratrol, we found that 140R rootstocks had the greatest change in resveratrol content after grafting, and resveratrol, as a phytochemical in the plant body, can improve the plant's resistance to disease, and it is also known as the main reason for the “French Paradox”, and it has the outstanding antimicrobial, anti-inflammatory, preventing heart disease and antioxidant and anticancer effects on the human body. Effectiveness. The content of resveratrol in grapes has become an important indicator for evaluating the quality of grapes. We thought this was a novel phenomenon, so we used it as a starting point for our experiment.
Comments 3: How many plants were analyzed for stilbenes? Usually, stilbenes can vary greatly among grape plants, so it is difficult to get a small SD, this is normal. In this case, I see a very small SD, this raises questions. It looks like several technical replicates from the same plant. I would like to clarify this issue.
Response 3:Thank you for pointing this out. In this study we set up a total of 3 replicates of 10 plants each, in order to minimize the error between plants, we measured the resveratrol content after sampling and mixing the 10 samples from each replicate, so the sd was smaller.
Comments 4: Figure 3. It is necessary to make the figures larger, since the legends are impossible to distinguish and it is difficult to evaluate the data. This remark applies to all figures in this article. Previously, in Figure 2, the authors used capital letters to denote figures, in this and subsequent figures, small letters are presented.
Response 4:Thanks for pointing this out, we have made changes to the graph in the article.
Comments 5: I do not understand why you provide data on scRNA and cnRNA in Table 1. To s
how that they do not exist?
Response 5:Thank you for pointing this out. We have deleted the scRNA and cnRNA data with zero counts as noted in Table 1, and have condensed the remaining data into Supplementary Table 1. We have also annotated the appearance of snoRNA, etc.
Comments 6: 2.6. Differential miRNA analysis. This should be very interesting data, but in fact you don't have it. In the text you describe this data poorly, and in the figure it is impossible to understand anything. This needs to be corrected.
Response 6:Thanks for pointing this out, I have re-described 2.6 of the analysis of the results of the article.
Comments 7: 2.8 RT-qPCR data. There is no statistical processing of the data at all. This is unacceptable. It is also necessary to describe the obtained results in detail. It is also necessary to use several housekeeping genes to normalize the data, this is the only way to reliably estimate the expression level.
Response 7:Thank you for pointing this out.We reanalyzed the data with ANOVA and modifications.
Reviewer 3 Report
Comments and Suggestions for Authors
Dear authors,
everything that is connected with grafting of grapevine is accurate in this modern world where breeding of grapevine is represented in all scientific conferences on a high level. Climatic changes requires new cultivars adaptible for new environnmental conditions. Wine sector is important on worldÅ› level, so it is necessary to give some data about how this process can be improved.
In the present article we can observe the resveratrol level that is important for grapevine and how miRNA can regulate the resveratrol content in the scion.
The article is good designed with appropiate tables, graphs and explanations. The results of morphology is good explained by picture. The sequencing is a hudge table (Table1.) but is should be written somehow, so this is good solution.
344-350 I think for grafting test part three replications with 10 plants is enough.
The literature used is accurate and up to date. Especially, the Li et all., are very deep in the resveratrol research of grape (the reference 3. and 4.)
Considering all mentioned, I recommend this article for publishing.
Regards
Author Response
Thank you very much for your scientific rigor in reviewing the manuscript.
The following are responses to the comments:
Comments 1: The article is good designed with appropiate tables, graphs and explanations. The results of morphology is good explained by picture. The sequencing is a hudge table (Table1.) but is should be written somehow, so this is good solution.
Response 1: Thank you for pointing this out. We agree with this comment. To better present our results, we have condensed the data in Table 1 and placed it in the Supplementary Data Table 1.
Comments 2: 344-350 I think for grafting test part three replications with 10 plants is enough.
Response 2: Thank you for pointing this out. Considering that stilbene is more variable between plants, the replicate tree chosen was 10 plants in order to minimize the error between plants.
Round 2
Reviewer 1 Report
Comments and Suggestions for Authors
Authors have adequately addressed the comments made by the reviewer in the revised version of the manuscript.
Author Response
Dear Reviewer
I am sincerely grateful that you took the time out of your busy schedule to review my paper. Your professional opinions and valuable suggestions played a crucial role in improving the quality of the paper. Your meticulous review and constructive feedback have enabled me to better refine and deepen the paper.
I am deeply appreciative of your support and recognition. Your acceptance means that my research findings have been recognized by an expert, which is a great encouragement to me.
Thank you again for your precious time and professional guidance.
Reviewer 2 Report
Comments and Suggestions for Authors
Thanks to the authors for the work done to improve the manuscript, but more could have been done. It has become better, but not enough to publish the work.
Comments 1: Accept
Comments 2: A controversial point about the stability of the cis-isomer, but overall the answer is clear.
Comments 3: Good
Comments 4: Figure 3 has become better, but there is Figure 6 (Chapter miRNA target gene analysis and functional annotations of the differentially expressed miR-NAs) and Figure 4 that are still of very poor quality.
Comments 5: Good
Comments 6: Accepted
Comments 7: Thanks for the improved statistics, but part of the issue with the need to use several housekeeping genes for data normalization remains open. Also, the description of the results remains at the same level.
Author Response
Dear Reviewer
I am sincerely grateful that you took the time out of your busy schedule to review my paper. Your professional opinions and valuable suggestions played a crucial role in improving the quality of the paper. Your meticulous review and constructive feedback have enabled me to better refine and deepen the paper.
The following are responses to the comments:
Comments 1: Figure 3 has become better, but there is Figure 6 (Chapter miRNA target gene analysis and functional annotations of the differentially expressed miR-NAs) and Figure 4 that are still of very poor quality.
Response 1:Thank you for pointing this out. We made a modification to Figure 4, Figure 6 is a combination of several images, resulting in poor image quality, we changed Figure 6A to Supplementary Figure 3, Figures 6B-C to Supplementary Figure 4, and Figures 6D-E to Figure 6A-B, to make the images as clear as possible.
Comments 2: Thanks for the improved statistics, but part of the issue with the need to use several housekeeping genes for data normalization remains open. Also, the description of the results remains at the same level.
Response 2:Thank you for pointing this out. We have modified the description of Figure 7 to analyze the contents of Figure 7. Regarding the fact that only U6 was selected as an internal reference because we chose multiple internal references to screen and determine in the early stage, in addition, we reviewed several articles in which quantitative analysis of miRNAs used one internal reference gene, such as (Pagliarani, Vitali et al. 2017, Niu, Li et al. 2019, Zhao, Xiao et al. 2022 , Ma Li 2023), etc., thank you very much for pointing this out, which is a very important help for our future research.
Reference
- Niu, C., H. Li, L. Jiang, M. Yan, C. Li, D. Geng, Y. Xie, Y. Yan, X. Shen, P. Chen, J. Dong, F. Ma and Q. Guan (2019). "Genome-wide identification of drought-responsive microRNAs in two sets ofMalus from interspecific hybrid progenies." Horticulture Research 6.
- Pagliarani, C., M. Vitali, M. Ferrero, N. Vitulo, M. Incarbone, C. Lovisolo, G. Valle and A. Schubert (2017). "The Accumulation of miRNAs Differentially Modulated by Drought Stress Is Affected by Grafting in Grapevine." Plant Physiology 173(4): 2180-2195.
- Zhao, W., W. Xiao, J. Sun, M. Chen, M. Ma, Y. Cao, W. Cen, R. Li and J. Luo (2022). "An Integration of MicroRNA and Transcriptome Sequencing Analysis Reveal Regulatory Roles of miRNAs in Response to Chilling Stress in Wild Rice." Plants-Basel 11(7).
- Ma L. miR393 mediates growth hormone-induced peach fruit softening. Master’s Thesis,Zhejiang university, Hangzhou, China, 2023.

Round 3
Reviewer 2 Report
Comments and Suggestions for Authors
Thank you for the work done. This version of the manuscript looks better.